# Genome-Wide Association Study Identifies Candidate Genes Associated with Feet and Leg Conformation Traits in Chinese Holstein Cattle

**DOI:** 10.3390/ani11082259

**Published:** 2021-07-30

**Authors:** Ismail Mohamed Abdalla, Xubin Lu, Mudasir Nazar, Abdelaziz Adam Idriss Arbab, Tianle Xu, Mohammed Husien Yousif, Yongjiang Mao, Zhangping Yang

**Affiliations:** 1College of Animal Science and Technology, Yangzhou University, Yangzhou 225009, China; ismailhmk@gmail.com (I.M.A.); dx120180094@yzu.edu.cn (X.L.); drmudasirnazar457@gmail.com (M.N.); arbabtor@yahoo.com (A.A.I.A.); cattle@yzu.edu.cn (Y.M.); 2Biomedical Research Institute, Darfur College, Nyala 63313, Sudan; 3Joint International Research Laboratory of Agriculture and Agri-Product Safety, Yangzhou University, Yangzhou 225009, China; tl-xu@outlook.com; 4Faculty of Animal Production, West Kordufan University, Alnuhud City 12942, Sudan; mohammedago@hotmail.com

**Keywords:** feet and leg, GWAS, FarmCPU, SNP, functional analysis, Chinese Holstein cow

## Abstract

**Simple Summary:**

Feet and leg problems are among the major reasons for dairy cows leaving the herd, as well as having direct association with production and reproduction efficiency, health (e.g., claw disorders and lameness) and welfare. Hence, understanding the genetic architecture underlying feet and conformation traits in dairy cattle offers new opportunities toward the genetic improvement and long-term selection. Through a genome-wide association study on Chinese Holstein cattle, we identified several candidate genes associated with feet and leg conformation traits. These results could provide useful information about the molecular breeding basis of feet and leg traits, thus improving the longevity and productivity of dairy cattle.

**Abstract:**

Feet and leg conformation traits are considered one of the most important economical traits in dairy cattle and have a great impact on the profitability of milk production. Therefore, identifying the single nucleotide polymorphisms (SNPs), genes and pathways analysis associated with these traits might contribute to the genomic selection and long-term plan selection for dairy cattle. We conducted genome-wide association studies (GWASs) using the fixed and random model circulating probability unification (FarmCPU) method to identify SNPs associated with bone quality, heel depth, rear leg side view and rear leg rear view of Chinese Holstein cows. Phenotypic measurements were collected from 1000 individuals of Chinese Holstein cattle and the GeneSeek Genomic Profiler Bovine 100 K SNP chip was utilized for individual genotyping. After quality control, 984 individual cows and 84,906 SNPs remained for GWAS work; as a result, we identified 20 significant SNPs after Bonferroni correction. Several candidate genes were identified within distances of 200 kb upstream or downstream to the significant SNPs, including *ADIPOR2, INPP4A, DNMT3A, ALDH1A2, PCDH7, XKR4* and *CADPS.* Further bioinformatics analyses showed 34 gene ontology terms and two signaling pathways were significantly enriched (*p* ≤ 0.05). Many terms and pathways are related to biological quality, metabolism and development processes; these identified SNPs and genes could provide useful information about the genetic architecture of feet and leg traits, thus improving the longevity and productivity of Chinese Holstein dairy cattle.

## 1. Introduction

Body conformation traits are considered economically important traits in dairy cattle [1]. Improving the accuracy of selection for body conformation traits would enhance the dairy industry as a whole and greatly affect the profitability of individual farms, which will automatically lead to the future profitability of the Chinese dairy industry. Thus, these traits can be used as an indirect predictor of economically important traits for an animal.

Adequate locomotion of an animal is directly associated with production and reproduction efficiency, as well as health and welfare. If an animal has feet and leg problems and bad mobility, it will be difficult to get access to feed and the milking parlor. Consequently, its longevity, welfare, productivity and reproductive performance will be affected [2] and, at most, culled before their third parity, due to such leg problems [3,4]. Several studies have reported significant genetic correlation between feet and leg conformation traits and their health problems (e.g., claw disorders and lameness), ranging between low, moderate and high, indicating that it is useful to use feet and leg conformation traits as an indirect selection criterion to control their health problems [5,6].

Over the last decades, genome-wide association studies (GWASs) have become a powerful tool in the search of potential genetic markers across the genome in dairy cattle, due to the availability of a high-density chip with single nucleotide polymorphisms (SNPs) for bovine. Phenotypic measurements and pedigree information, along with sequence variations (mainly single nucleotide polymorphisms, SNPs) in the whole genome, serve as the basic pillars for genome-wide association studies (GWAS). Altogether, they identify genes or regulatory elements necessary for each trait of interest [7].

In the last decade, various GWAS studies have been under taken on Chinese Holstein cattle, involving milk production traits [8,9] milk composition traits [10,11], fatty acids [12,13], protein [14], body conformation traits [15,16], body size [17], pigmentation [18], fertility [19], concentration of albumin in colostrum and serum [20] and mastitis [21,22].

Regarding the traits under investigation in this paper, some studies have been conducted which identified many QTL and genes associated with feet and legs conformation traits in dairy cattle using the GWAS method [16,23,24,25].

Therefore, the objectives, in our study, were to detect significant single nucleotide polymorphisms (SNPs) associated with feet and legs conformation traits (heel depth, Bone quality, rear leg side view and rear leg rear view) in Chinese Holstein cattle using the genome-wide association (GWAS) approach and using the SNPs positions to identify candidate genes and pathways that may influence these traits. In addition, we will use the identified genes to conduct some bioinformatics analyses, such as gene ontology, KEGG pathway and gene network analysis for further analysis.

## 2. Materials and Methods

### 2.1. Ethics Statement

All procedures for collecting hair follicle samples and measuring phenotypic traits were carried out strictly in accordance with the guidelines proposed by the China Council on Animal Care and Ministry of Agriculture of the People’s Republic of China. The study was also approved by the Institutional Animal Care and Use Committee of School of the Yangzhou University Animal Experiments Ethics Committee (License Number: SYXK (Su) IACUC 2012-0029), Yangzhou University.

### 2.2. Body Measurements and DNA Samples Collection 

The experimental population consisted of 1000 Chinese Holstein cows raised on four farms, (199 individuals from Sihong Farm, 214 from Xuyi Farm, 224 from Xuzhou Farm and 363 from Huaxia Farm). All these farms are located in the northern part of Jiangsu province, China. Phenotypic descriptive traits (BQ, bone quality; HD, heel depth; RLSV, rear leg side view; RLRV, rear leg rear view) were measured individually and scored on a 1–9 scale score according to the National Standards of People’s Republic of China Code of practice of type classification in Chinese Holstein (GBT35568-2017). The measurement of traits for each cow were performed by three trained technicians and the average of the measurements taken by the different professionals was used as the phenotype for each trait to ensure the accuracy of the data. For the genotyping analysis, hair follicle samples were collected individually and stored in special paper envelopes for each cow to prevent it from environmental and DNA contamination; each paper envelope must contain enough hair samples, not less than 50 hairs.

### 2.3. Phenotypic and Genetic Parameters 

Descriptive statistics and pairwise Pearson correlation coefficients of phenotypic traits were determined using the computer-based software IBM-SPSS, version 25.

Genetic analysis was carried out using the Derivative-free approach to MUltivariate analysis (DMU) software [26] to estimate heritability and genetic correlation between pairs of traits with the animal model as follows:$$y_{ijklm}=u+Herd_{i}+Year_{j}+Season_{k}+Parity_{1}+a_{m}+e_{ijklm}$$
where ***y_ijklm_*** is the phenotype in the ***j^th^*** year, ***k^th^*** season and ***l^th^*** parity of the ***m^th^*** individual from the ***i^th^*** herd; ***u*** is overall mean of the population, ***Herd_i_*** is the herd effect according to a cow’s origin from one of the four herds; ***Year_j_*** is the ***j^th^*** year effect, ***Season_k_*** is the ***k^th^*** season effect and parity is the effect of ***l^th^*** parity; ***a*** is the additive effect of the ***m^th^*** individual, which was evaluated by the pedigree information, and ***e*** is the residual in the ***j^th^*** year, ***k^th^*** season and ***l^th^*** parity of the ***m^th^*** individual from the ***i^th^*** herd. All effects were treated as random except the overall mean. The pedigree of the cows could be traced back at least three generations (2009–2020), the parities of cows were between 1 and 4 and four seasons of were defined, December–February, March–May, June–August and September–November.

### 2.4. Genotyping and Quality Control

Each individual from the experimental population was genotyped using the GeneSeek Genomic Profiler Bovine 100 k SNP chip (Neogen Corporation, http://www.neogenchina.com.cn/ (accessed on 28 June 2020)) based on ARC-UCD1.2/bosTau9 as the genome reference. Genomic DNA was extracted from hair follicle samples. The GeneSeek Genomic Profiler Bovine 100 K SNP chip containing 100,000 SNPs was utilized for individual genotyping. 

Quality control was conducted by using the Plink 1.90 software [27] to remove the markers which did not comply with the following criteria: (1) individual call rate lower than 95%; (2) genotype call rate of a single SNP lower than 90%; (3) minor allele frequency (MAF) > 0.05; (4) deviated from Hardy–Weinberg equilibrium (*p* < 1.0 × 10^−6^). After quality control, 984 cows and 84,406 markers remained for further analyses (Appendix A).

Marker intervals and linkage disequilibrium (LD) were calculated to estimate R square for all markers and plotted the marker distribution as showed in Figure 1d. LD decay fell off quickly within 100 kb physical distance, then decreased slowly afterward.

### 2.5. Population Structure Analysis

We used the Plink 1.90 software [27] to implement principal component analysis (PCA) on 1000 cows genotyping with 84,406 SNPs in Chinese Holstein cattle herds raised at four breeding herds to investigate the population structure and PCA was plotted using the ggplot package In R 4.0.2. The admixture software [28] was also used to study the population structure and correct population stratification.

### 2.6. Genome-Wide Association Studies

In this study, the fixed and random model circulating probability unification (FarmCPU) method was used to carry out the genome-wide association analysis [29]. The FarmCPU method uses a fixed effect model and a random effect model iteratively. In GWASs, population stratification is the main cause of false positive correlations. Therefore, the fixed effect model tests SNPs one at a time. The significant SNPs are evaluated in the random effect model and the validated SNPs are fitted as covariates in the fixed effect model to control population structure. The model can be written as follows:(1)$$y_{i\;}=M_{i1}b_{1}+M_{i2}b_{2}+\cdots +M_{it}b_{t\;}+S_{ij}d_{i}+e_{i}$$
where ***y_i_*** is the observation of the ***i^th^*** individual; ***M*_*i*1_**, ***M*_*i*2_**…***M_it_*** are the genotypes of t pseudo QTNs, initiated as an empty set; ***b*_1_**, ***b*_2_**, …, ***b_t_*** are the corresponding effects of the pseudo QTNs; ***S_ij_*** is the genotype of the ***i^th^*** individual and ***j^th^*** genetic marker; ***d_j_*** is the corresponding effect of the ***j^th^*** genetic marker; ***e_i_*** is the residual having a distribution with zero mean and variance of ***σ*^2^*_e_***.

After substitution, every marker has its own *p* value. The *p* values and the associated marker map are used to update the selection of pseudo QTNs using the SUPER algorithm (Settlement of MLM Under Progressively Exclusive Relationship) [30] in a REM as follows: (2)$$y_{i\;}=u_{i}+e_{i}$$
where ***y_i_*** and ***e_i_*** are the same as in Equation (1) and ***u_i_*** is the total genetic effect of the ***i*th** individual.

The total type 1 error (false positive) rate was controlled at 5% and the significance threshold of the GWAS was determined according to this formula (0.05/Nsnp), where Nsnp is the number of SNPs remaining after quality control [31]. Subsequently, the significant threshold for the GWAS was 5.9 × 10^−7^ (0.05/84,406) after Bonferroni correction.

Quantile–quantile (Q–Q) and Manhattan plots were drawn using the CMplot package in the R 3.1.1 software [32]. 

### 2.7. Gene Identification

Genomic regions and candidate genes were recognized on the livestock using a genome browser (UCSC) through an Asian server for cow assembly April 2018 (ARC-UCD1.2/bosTau9). https://genomeasia.ucsc.edu/cgibin/hgGateway?redirect=manual&source=www.genome.ucsc.edu (accessed on 25 May 2021) and full NCBI (the National Center for Biotechnology Information Gene) database (http://www.ncbi.nlm.nih.gov/gene/ (accessed on 25 May 2021)). 

### 2.8. Function and Pathway Enrichment and Network Analysis 

In our study, we submitted the candidate genes obtained by GWAS into the Database for Annotation, Visualization and Integrated Discovery (DAVID) [33] for the Gene Ontology (GO) terms [34] and Kyoto Encyclopedia of Genes and Genomes (KEGG) pathway analysis [35]. The statistically significant *p* value for functional analysis and pathway analysis was defined at *p* ≤ 0.05. Protein–protein interactions among genes were performed using the online Search Tool for the Retrieval of Interacting Genes (STRING) database v11.0 [36] with the Cytoscape software.v3.8.2 to visualize the resultant PPI network.

## 3. Results

### 3.1. Descriptive Statistics and Heritability Estimation of Feet and Legs Traits

The feet and leg traits for the 1000 Chinese Holstein cows from four farms measured in this study included heel depth (HD), bone quality (BQ), rear leg side view (RLSV) and the rear leg rear view (RLRV). The descriptive statistics (mean, maximum and minimum values and standard deviation) of phenotypic measurements for these traits are showed in Table 1, where the mean of the HD trait had the greater score (7.04) and the RLSV trait revealed the lowest score, with 3.92. The pairwise genetic and Pearson phenotypic correlation between the traits are provided in Table 2. BQ phenotypically positively correlated with RLSV, and the other traits were low or moderate negatively correlated with RLSV. However, BQ was genetically positively correlated with HD, RLSV and RLRV. HD was negatively correlated with all traits, both genetically and phenotypically, except BQ had a highly positive genetic correlation. RLSV was highly positively genetically correlated with BQ, while it was negative with HD and RLRV, whereas it had a negative phenotypic correlation with BQ, HD and RLRV. Estimates of the heritability results were 0.15, 0.05, 0.17 and 0.15 for heel depth, bone quality, rear leg side view and rear leg rear view, respectively (Table 2).

### 3.2. Information of SNPs 

After quality control, 984 individual cows and 84,406 SNPs were used to conduct the GWAS. The filtered SNPs were distributed on all 29 chromosomes. Chromosome 1 has shown the greatest number of SNPs, whereas chromosome 25 contained the fewest. The minor allele frequency (MAF) for all SNPs was re-calculated after quality control; only MAF above 5% remained. The LD decay line dramatically decreased at the beginning then tended to be slow after 100 kb distance. (Figure 1).

### 3.3. Population Stratification 

Principal component analysis (PCA) and the ADMIXTURE program were used to visualize family structure in this study. The results reveal all individuals were grouped into two unequal sized clusters, as shown in Figure 2a,b; as for the admixture population, using ADMIXURE with k value ranging from 1 to 7 (Figure 2c), based on the cross-validation errors, K = 4 was identified to be the optimal number of genetic clusters defining the population structure among the four herds of Chinese Holstein cattle. Also, the results show that the first two principal components were about 21% of the variation (Figure 2a), and they were fitted as covariate variables in the association analysis using the FarmCPU model. Population stratification based on the PCA analysis results was considered and incorporated into the mixed linear model.

### 3.4. GWAS Results

The FarmCPU model was used to conduct the genome-wide association analysis in the present study. In Figure 3, The quantile–quantile (Q–Q) plots illustrated the model used in this study for GWAS analysis was reasonable. The lambda (inflation factor (λ)) was close to 1 (0.9 < 1.01) and the point at the upper right corner of (Q–Q) plots are the significant markers associated with the traits under study (Figure 3); therefore, the population stratification was adequately controlled. Meanwhile, Manhattan plots are used to visualize GWAS significance level (−log10 of *p* value of each SNP) by chromosome location (Figure 4).

In the present study, 20 SNPs (Table 3) were past the threshold and significantly associated with four traits of feet and legs (heel depth, bone quality, rear leg side view and rear leg rear view).

For the bone quality trait, five SNPs were detected on the chromosomes Chr8 (Hapmap54208-rs29015846-rs29015846), Chr4 (BovineHD0400024774-rs133088614), Chr17 (ARS-BFGL-NGS-81828-rs41845981), Chr14 (BovineHD1400007035-rs136017102) and Chr22 (BovineHD2200011035-rs110949452), while six SNPs associated with heel depth trait were detected on the chromosomes Chr20 (BovineHD2000006450-rs137022628 and ARS-BFGL-NGS-116157-rs109601642), Chr23 (ARS-BFGL-BAC-36389-rs109652453), Chr27 (BovineHD2700002886-rs42110372), Chr15 (ARS-BFGL-NGS-73835-rs41577664) and Chr14 (BovineHD1400023839- rs134726669). Moreover, three significantly SNPs associated with rear leg rear view were detected on chromosomes Chr3 (BovineHD0300015960-rs134130409), Chr20 (BovineHD4100014792-rs134139959) and Chr1 (Hapmap48798-BTA-51401-rs41638134). In addition, six SNPs associated with rear leg side view were detected on the chromosomes Chr5 (ARS-BFGL-NGS-91167-rs41565304), Chr3 (BovineHD0300022142-rs43350216), Chr11 (BovineHD1100001372-rs43656945 and BovineHD1100021155-rs136593856), Chr10 (BTB-01677645-rs42791722) and Chr6 (BTB-01518251-rs42639670) (Table 3).

### 3.5. Identified Candidate Genes 

The results of the linkage disequilibrium (LD) analysis indicate the LD decay (R^2^), as showed in Figure 1d, tends to be stable when the distance is 200 kb. Therefore, genes located within this region (200 kb) of the significant SNP are defined as candidate genes.

Among all the 20 significant SNPs (Table 3), 8 of them (ARS-BFGL-NGS-81828-rs41845981, BovineHD2200011035-rs110949452, ARS-BFGL-BAC-36389-rs109652453, BovineHD4100014792-rs134139959, ARS-BFGL-NGS-91167-rs41565304, BovineHD1100001372-rs43656945, BovineHD1100021155-rs136593856 and BTB-01677645: rs42791722) are located within genes, namely, DNA polymerase epsilon (*POLE*), calcium dependent secretion activator (*CADPS*), synaptonemal complex protein 2 like (*SYCP2L*), F-box and leucine rich repeat protein 7 (*FBXL7*), adiponectin receptor 2 (*ADIPOR2*), inositol polyphosphate-4-phosphatase type I A (*INPP4A*), DNA methyltransferase 3 alpha (*DNMT3A*) and aldehyde dehydrogenase 1 family member A2 (*ALDH1A2*), respectively (Table 3).

Whilst the SNP (Hapmap54208-rs29015846-rs29015846) on Chr8 is located near (100 kb) the U6 spliceosomal RNA gene (*LOC112447952*), the SNP (BovineHD0400024774: rs133088614) on Chr4 is located close (100 kb) to the transmembrane protein 229A gene (TMEM229A) and the SNP (BovineHD1400007035-rs136017102) on Chr14 is located close (1 kb) to the XK related 4 gene (*XKR4*). The SNP (BovineHD2200011035-rs110949452) on Chr20 is located close (50 kb) to actin beta like 2 gene (*ACTBL2*). The SNP (BovineHD2700002886-rs42110372) on Chr27 is located close (100 kb) to the small nucleolar RNA SNORD22 gene (*LOC112444670*). The SNP (ARS-BFGL-NGS-116157-rs109601642) on Chr20 is located close (100 kb) to gene *LOC101907219*. The SNP (ARS-BFGL-NGS-73835-rs41577664) on Chr15 is located close (200 bp) to gene *LOC112441589*. The SNP (BovineHD1400023839-rs134726669) on Chr14 is located close (200 bp) to the mitochondrial ribosomal protein L13 gene (*MRPL13*). The SNP (BovineHD0300015960-rs134130409) on Chr3 is located close (100 kb) to BarH like homeobox 2 gene (*BARHL2*). The SNP (Hapmap48798-BTA-51401-rs41638134) on Chr1 is located close (2 kb) to gene *LOC107132214*. The SNP (BovineHD0300022142) on Chr3 was not harboring any gene within 200 kb. Moreover, the SNP (BTB-01518251-rs42639670) on Chr6 is located close (200 kb) to the gene protocadherin 7 (*PCDH7*).

### 3.6. Functional Analysis

Using the UCSC Genome browser and NCBI database through the Asian server for cow assembly April 2018 (ARC-UCD1.2/bosTau9), 105 genes were obtained within the region of 200 kb up/downstream of the significant SNPs for the feet and legs traits; these genes were used for further enrichment and pathway analysis (gene ontology and KEGG). Gene ontology enrichment analysis revealed 34 GO terms were significantly enriched (*p* < 0.05), which consist of 18 biological process terms, 1 cellular component and 15 molecular function terms (Appendix A and Figure 5).

The KEGG pathways analysis revealed that five pathways (bta00531:Glycosaminoglycan degradation involves 3 genes and the bta01100:Metabolic pathway consists of 12 genes) were enriched (*p* ≤ 0.05), whereas the other three pathways (bta04020:Calcium signaling pathway, bta05414:Dilated cardiomyopathy and bta04916:Melanogenesis) did not reach the significant *p* value (*p* ≤ 0.05) (Table 4 and Figure 5).

The STRING database was used to conduct protein–protein interaction network (PPIN) analysis using all the genes previously used in the functional analysis, to identify the interactions between these genes. Figure 6 shows that there is several interacting between genes (containing 78 nodes connected via 139 edges); the proportional interaction strength between these genes was shown by the intensity of staining between the lines that linked one gene to another (Figure 6).

## 4. Discussion

The esitmated genetic correlations in our study ranged between −0.35 and 0.84 (Table 2); these results are in agreement with the study by Olasege et al. [37], who reported that between moderate negative and strong postive genetic correlation existed among feet and leg traits, ranging from −0.35 (between rear leg view and set of rear legs) to 0.74 (between foot angle and bone quality) in Chinese Holstein cattle. Among the feet and leg traits, the strongest genetic correlation was found between RLSV and BQ (0.84) (Table 2); if two traits have a high and postive genetic correlation, it implies that most QTL affect them both in the same direction [38]. For intance, most SNPs with significant influences affect weight and height in the same direction, thus helping to explain the known high genetic correlation between these two traits [39].

Population stratification or cryptic relatedness are considered major challenges that face gnome-wide association analysis. Thus, the presence of population stratification can cause spurious association due to systematic ancestry differences [40] that can lead to false positives in GWAS results [41]; therefore, stratification must be well controlled across the experimental population to maximize power to detect true associations [42,43].

There are several methods to correct population stratification; carefully choosing the statistical model can be a useful method to correct and minimize the chances of type 1 error (false positive associations) [44]. 

To eliminate false positives, the most effective strategy is either (1) fitting population structure as covariates in a General Linear Model (GLM) [45] or (2) fitting both population structure and each individual’s total genetic effect as covariates in a Mixed Linear Model (MLM) [46] to make adjustments for testing markers. As MLM can miss some potentially important discoveries and lead to false negatives, due to the confounding among population structures, quantitative trait nucleotides (QTNs) and kinship. Therefore, in the present study, we conducted Fixed and random model Circulating Probability Unification (FarmCPU) model analysis because of its advantages to completely control false positives, eliminate confounding and improve computational efficiency by using the fixed effect model and random effect model iteratively [29].

The inflation factor (λ) should be close to 1, after correcting the population stratification [45]. In the present study, the Q–Q plot in Figure 3 shows that the deviation of the observed value from the expected value is near to 1 and the inflation factor (λ) is 0.9 < 1.01, both indicating that the population stratification was properly corrected by using an appropriate model.

The principal component analysis (PCA) method [47] and ADMIXTURE program analysis [28] are able to detect the population structure by classifying individuals into grouped ancestry based on their genetic makeup. Figure 2a shows all the experimental population was clustered into two groups (one large and one small). This indicates that each group clustered closely together has a genetic relationship. The division in two groups might be attributed to using Holstein semen from different countries or the cows in these farms may contain blood from other breeds, as it is known one of the requirements for registering Holstein cattle in China is that the cattle have at least 87.5 percent blood of Holstein (Chinese Holstein, GB/T 3157 2008). 

In the present study, genome-wide association analysis identified 20 significant SNPs associated with four traits belonging to feet and legs trait in Chinese Holstein cattle using the FarmCPU model; among them, the most three significant SNPs were BovineHD2000006450-rs137022628, located near the *ACTBL2* gene, ARS-BFGL-NGS-91167-rs41565304, located within the ADIPOR2 gene and no genes were found within 200 kb upstream or downstream of the BovineHD0300022142-rs43350216 SNP.

Until now, only one report has discussed the functional role of the *ACTBL2* gene. Hödebeck et al. [48] reported that the silencing of *ACTBL2* leads to diminished motility of human arterial smooth muscle cells. These authors also demonstrated that the expression of ACTBL2 in smooth muscle cells under stretch conditions depends on the nuclear factor 5 of activated T-cells (NFAT5).

Adiponectin receptor 2 (*ADIPOR2*) is a member of the adipocytokines gene family. Generally, adiponectin receptor genes play an important role in bone and whole-body energy homeostasis that could be induced by activation of the AMPK signaling pathway; *ADIPOR1* and *ADIPOR2* also play a crucial role in metabolic pathways, which lead to the regulation of lipid and glucose metabolism, oxidative stress and inflammation [49]. Ouyang et al. [50] reported *ADIPOR2* has a role in intramuscular fat content in the longissimus dorsi (LD) muscle of different pig breeds. Lewis et al. [49] discussed in detail the role of adiponectin signaling in bone homeostasis and its effect on the lifestyle of human healthy aging and disease. Moreover, *ADIPOR2* is considered to be one of the genes associated with conformation and reproductive performance traits (hip height, rump length, calving ease, height, ovulation, type and rump angle) in one of the South African cattle breeds [51].

In addition to the *ADIPOR2* gene, our study identified other genes that were significantly associated with rear-leg side view traits (*INPP4A*, *DNMT3A*, *ALDH1A2* and *PCDH7*).

Some studies concerning the inositol polyphosphate-4-phosphatase type I A gene (INPP4A) reported it to have an association with the thoracic vertebrae number in sheep [52] and a role in regulating the normal functioning of the spinal cord neurons [53]. Additionally, the metabolic process [54] also presented different isoforms in transcriptomic investigation of meat tenderness in two Italian cattle breeds [55].

The DNA methyltransferase 3 alpha gene (*DNMT3A*) was associated to obese adipose tissue in transgenic mice [56] and significantly associated with beef quality traits [57]. Meanwhile, the aldehyde dehydrogenase 1 family member A2 gene (*ALDH1A2*) was associated with the growth trait in the Blanco Orejinegro cattle breed from Colombia [58], as well as intramuscular fatty acid composition in rabbits [59]. *ALDH1A2* was also associated with lactation persistency in Canadian Holstein cattle [60], carcass traits in Chinese Simmental beef cattle [61] and litter traits in Landrace and Large White pigs [62]. Moreover, *ALDH1A2* has been reported to play critical roles in the synthesis of retinoic acid, the active derivative of vitamin A, which is important for limb and organ development [63].

The protocadherin 7 gene (*PCDH7*) might be related to the body size trait of Chinese Simmental beef cattle [64], residual feed intake in Nelore cattle steers [65], feed efficiency trait in pig [66], internal organ traits in chickens [67] and sperm traits in Assaf rams [68].

The BovineHD4100014792-rs134139959 SNP located within F-box and leucine rich repeat protein 7 gene (*FBXL7*) was associated with the rear leg rear view trait in the present study, but, in previous studies, was associated with body length in water buffaloes (*Bubalus bubalis*) [69], clinical mastitis in first lactating US Holstein dairy cattle [70], subclinical ketosis in second and later lactations in Canadian Holstein dairy cattle [71] and also litter traits in Landrace and Large White pigs [62].

Furthermore, the results indicate the *TMEM229A*, *POLE*, *XKR4* and *CADPS* genes were harboring significant SNPs correlated to the bone quality trait. In relation to the BovineHD0400024774-rs133088614 SNP, located close to the transmembrane protein 229A gene (*TMEM229A*), Juan Carlos [72] described the polymorphism of this gene as having a great effect on dairy yield per lactation in Colombian Holstein cattle. There are no further publications concerning this gene in cattle.

Functional analyses of list candidate genes showed a DNA polymerase epsilon (*POLE*) gene potentially associated with reproductive traits [73]; they might be implicated in milk yield regulation by performing certain biological functions [74], while our findings suggest that *POLE* is associated with the bone quality trait.

The BovineHD1400007035-rs136017102 SNP on Chr14, located close to the *XKR4* gene, in previous studies, was reported to be associated with feed intake and growth phenotypes in cattle [75], residual feed intake in Australian Angus cattle [76], carcass weight (CWT) and eye muscle area (EMA) in Korean Hanwoo cattle [77], carcass trait [78], fat deposition in genome-wide association study for carcass trait in Brazilian Nellore cattle [79] and subcutaneous rump fat thickness in indicine and composite cattle [80].

The significant SNP BovineHD2200011035-rs110949452 on Chr22 is positioned within the calcium dependent secretion activator gene (*CADPS*); this gene is a member of the Ca^+2^-dependent activator for the secretion protein family [81]. Interestingly, our study detected this gene is associated with the bone quality trait, while Vargas et al. [25] annotated the same gene within the top windows related to the overall quality of feet and leg in Nellore cattle. Kim et al. [82] reported this gene as selection signatures in Korean Brindle Hanwoo cattle obtained from genome-wide SNP analysis. Another study in two pig breeds identified the *CADPS* gene as associated with body weight [83].

In the present study, we found the ARS-BFGL-BAC-36389-rs109652453 SNP, which is associated with heel depth trait, to be located within synaptonemal complex protein 2, such as *SYCP2L*. Previously, most *SYCP2L* gene studies related to fertility. He et al. [84] reported a novel gene responsible for human premature ovarian insufficiency. For instance, *SYCP2L* was demonstrated to play a significant role in the survival of oocytes [85]. This gene is a paralogue of the synaptonemal complex protein gene (*SYCP2*), regulating females’ reproductive aging. Therefore, it was suggested to have an association with the age of natural menopause in humans [85]. In addition, this gene is involved in the goat fecundity trait of Chinese Guangfeng goat [86].

Gene ontology enrichment analysis and KEGG pathways, in our study, revealed a number of GO terms and KEGG pathways. These involved significant genes related to the traits under examination, for example, the regulation of biological quality term (GO:0065008) in the biological process category, which has 22 genes (Figure 5 and Appendix A); among them, the *CADPS*, *BARHL2*, *ADIPOR2* and *ALDH1A2* genes were considered as nearest genes to our significant SNPs. According to the Mouse Genome Database (MGD) http://www.informatics.jax.org/ (accessed on 25 May 2021), this term and its sub-term (homeostatic process; GO:0042592) are related to any process that modulates a qualitative or quantitative trait of a biological quality, such as size, mass, shape, color, etc. As discussed earlier, some of the candidate genes involved in this term might play an important role in the biological regulation of body conformation traits and, subsequently, contribute to feet and legs traits.

In addition, the *LOC520336*, *MGAT4A*, *A4GNT*, *GCNT2* and *GALNT9* genes were clustered in certain biological processes and molecular functions. The terms (including carbohydrate metabolic process, GO:0005975, macromolecule glycosylation, GO:0043413, protein glycosylation, GO:0006486, glycosylation, GO:0070085, glycoprotein biosynthetic process, GO:0009101, glycoprotein metabolic process, GO:0009100, acetylglucosaminyltransferase activity, GO:0008375, UDP-glycosyltransferase activity, GO:0008194, transferase activity, transferring glycosyl groups, GO:0016757, and transferase activity, transferring hexosyl groups, GO:0016758) in our study revealed to be related to metabolic processes.

Moreover, there were two terms (embryonic forelimb morphogenesis, GO:0035115, and forelimb morphogenesis, GO:0035136) that were related to the limb development term. 

On another hand, the *P2RX3* and *P2RX2* genes were noticed in many biological process terms, such as urinary bladder smooth muscle contraction, GO:0014832, urinary tract smooth muscle contraction, GO:0014848, response to organophosphorus, GO:0046683, response to purine-containing compound, GO:0014074, peristalsis, GO:0022803, and in the molecular function category, such as channel activity (GO:0015267) and terms related with it (cation channel activity, gated channel activity, ATP-gated ion channel activity, extracellular ATP-gated cation channel activity and substrate-specific channel activity), and passive transmembrane transporter activity, GO:0022803. The *P2RX3* and *P2RX2* genes are subunits of the *P2X* gene family. *P2X* receptors are non-selective cation channels gated by extracellular ATP and exhibit relatively high Ca^2+^ permeability [87]; the *P2rx2* receptor mainly functions in sensory neurons, neuromuscular junction formation, whereas *P2RX3*, with multiple copies, were found in bony fishes [88].

The top two KEGG analysis pathways were bta00531: Glycosaminoglycan degradation, which has three genes, and bta01100: Metabolic pathways, which comprise 12 genes.

The functional analysis yielded many terms and pathways related to biological quality, metabolism and development. Therefore, it is reasonable to presume that all significant SNPs and candidate genes might be associated with feet and legs conformation.

## 5. Conclusions

In conclusion, this study identified 20 significant SNPs associated with feet and legs traits (bone quality, heel depth, rear leg side view and rear leg rear view) in Chinese Holstein cattle. Serval genes harbor SNPs (*ADIPOR2*, *INPP4A*, *DNMT3A*, *ALDH1A2*, *PCDH7*, *XKR4* and *CADPS*), identified mainly to participate in biological quality, metabolism and development processes. Our findings provide useful information to understand genetic architecture and give some fundamentals for molecular-based breeding, leading to genetic improvement programs on feet and leg traits. Further investigations to validate the biological functions of these genes are recommended.

## Figures and Tables

**Figure 1 animals-11-02259-f001:**
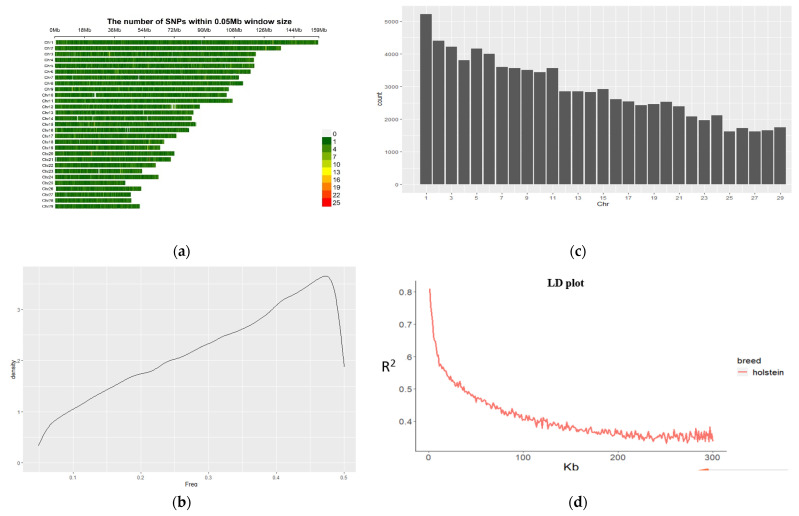
Properties of single nucleotide polymorphisms (SNPs). In total, 1000 individuals were genotyped by the GeneSeek Genomic Profiler Bovine 100 k bead chip; 84,609 SNPs and 984 cattle passed filters and quality control. Marker distributions are displayed in a heatmap on 29 chromosomes by minor allele frequency (MAF) (**a**). MAF was re-calculated after quality control. Therefore, some SNPs remained with MAFs lartherthan 0.05, as shown by the histogram (**b**). Marker density is displayed by the histogram on chromosome 29 (**c**). LD decay is shown by scatter plot according to pairwise distance and trend as a red line (**d**).

**Figure 2 animals-11-02259-f002:**
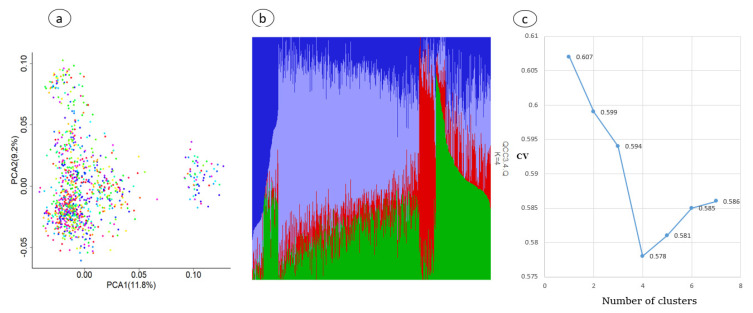
Population structure for experimental population (1000 cows) raised at four farms. (**a**) Principal component analysis. (**b**) Admixture analysis. (**c**) Cross-validation errors across 21 ancestral genetic clusters.

**Figure 3 animals-11-02259-f003:**
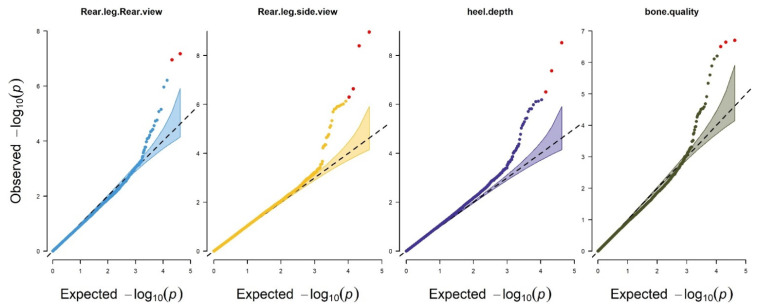
Quantile–quantile (Q–Q) from GWAS for feet and leg traits in Chinese Holstein cattle. Q–Q plot showing the late separation between observed and expected values. The red lines indicate the null hypothesis of no true association. Deviation from the expected *p* value distribution is evident only in the tail area for each trait, indicating that population stratification was properly controlled.

**Figure 4 animals-11-02259-f004:**
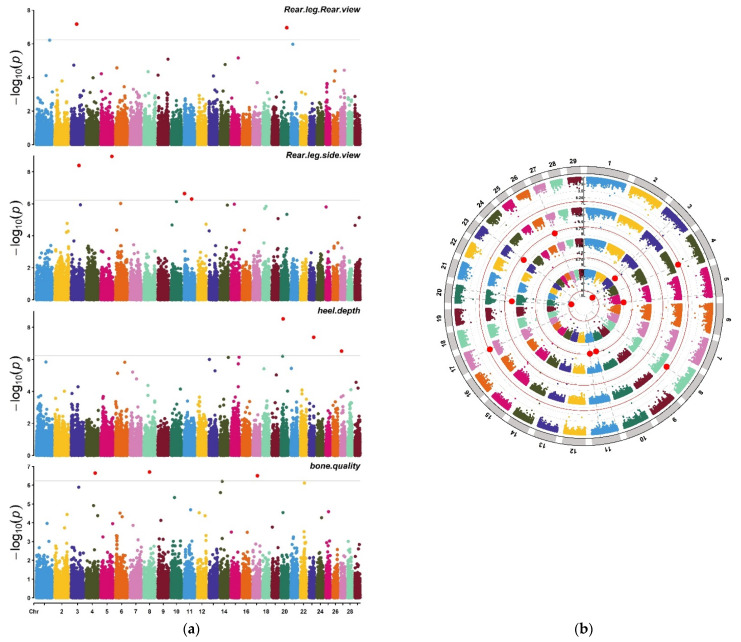
Manhattan from GWAS for feet and leg traits in Chinese Holstein cattle. Manhattan plot in which the genomic coordinates of SNPs are displayed along the horizontal axis, the negative logarithm of the association *p* value for each SNP is displayed on the vertical axis and the green line indicates the significance threshold level after Bonferroni correction (**a**). Circular Manhattan plot, the four feet and leg conformation traits are plotted from the outside to the inside, respectively (**b**).

**Figure 5 animals-11-02259-f005:**
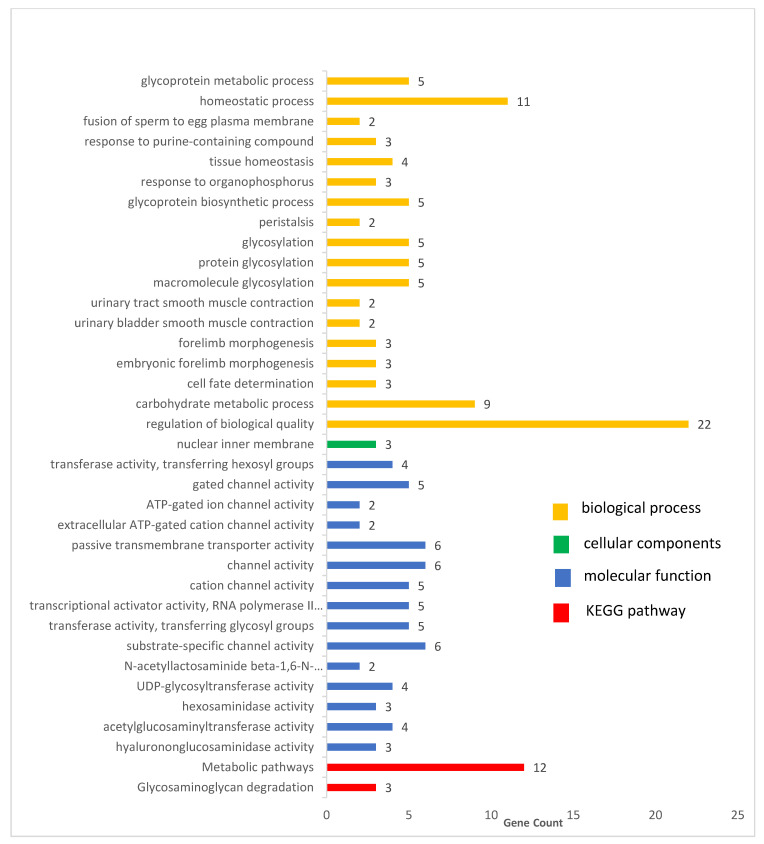
Significant GO terms (biological process, cellular component and molecular function) and KEGG pathways of candidate genes related to feet and leg traits.

**Figure 6 animals-11-02259-f006:**
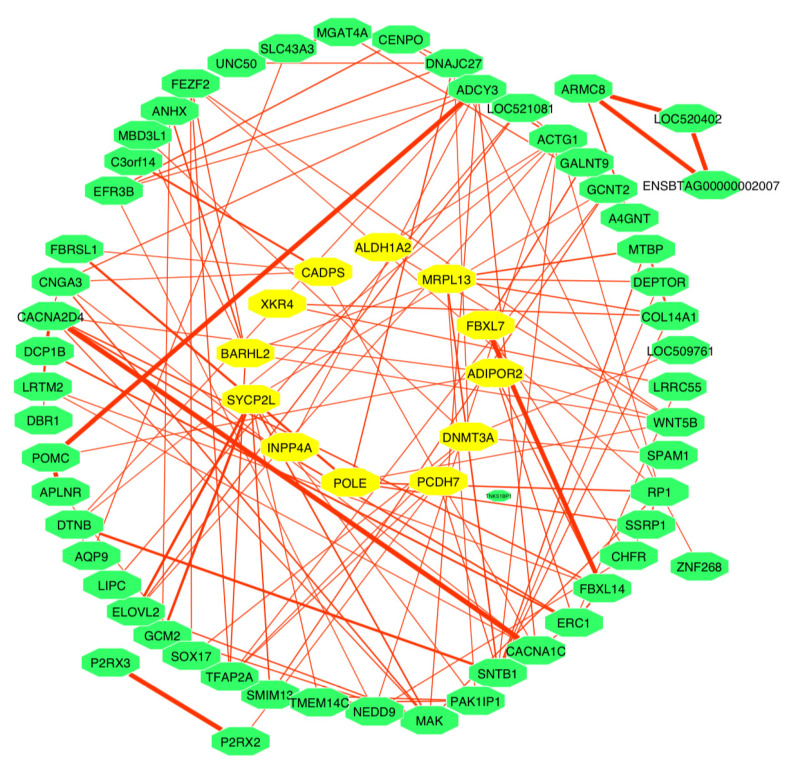
Gene interaction network for genes associated with feet and leg traits in Chinese Holstein cattle. (The nodes in yellow represent the significant candidate genes and their interacting partners).

**Table 1 animals-11-02259-t001:** Descriptive statistics of feet and leg conformation traits measurements.

Traits	Mean	Std. Error	Minimum	Maximum	Std. Deviation
HD	7.04	0.051	2.00	9.00	1.61
BQ	6.03	0.028	2.00	9.00	0.87
RLSV	3.92	0.048	1.00	9.00	1.50
RLRV	5.60	0.045	2.00	9.00	1.57

HD, heel depth; BQ, bone quality; RLSV, rear leg side view; RLRV, rear leg rear view.

**Table 2 animals-11-02259-t002:** Genetic (upper diagonal), Pearson phenotypic (lower diagonal) correlations and heritability (grey diagonal line) for feet and leg traits measurements.

Traits	HD	BQ	RLSV	RLRV
HD	0.15	0.67	−0.03	−0.36
BQ	−0.08 **	0.05	0.84	0.10
RLSV	−0.07 *	−0.08 *	0.17	−0.09
RLRV	−0.09 **	0.1 **	−0.39 **	0.15

HD, heel depth; BQ, bone quality; RLSV, rear leg side view; RLRV, rear leg rear view. The upper subscript * and ** represent significant correlation at the 0.05 and 0.01, respectively.

**Table 3 animals-11-02259-t003:** Genome-wide significant SNP associated with feet and leg traits and nearest candidate genes.

Traits	SNP Name	rs. SNP Name	Chr.	Position	MAF	*p* Value	Nearest Gene Name	Distance
Bone quality	Hapmap54208-rs29015846	rs29015846	8	55190206	0.29	1.99 × 10^−7^	*LOC112447952*	100 kb
	BovineHD0400024774	rs133088614	4	88636425	0.46	2.25 × 10^−7^	*TMEM229A*	100 kb
	ARS-BFGL-NGS-81828	rs41845981	17	44543822	0.35	3.14 × 10^−7^	*POLE*	within
	BovineHD1400007035	rs136017102	14	22608072	0.13	6.22 × 10^−7^	*XKR4*	50 kb
	BovineHD2200011035	rs110949452	22	38556301	0.34	7.67 × 10^−7^	*CADPS*	within
Heel depth	BovineHD2000006450	rs137022628	20	21510411	0.27	3.03 × 10^−9^	*ACTBL2*	200 kb
	ARS-BFGL-BAC-36389	rs109652453	23	45295465	0.37	4.22 × 10^−8^	*SYCP2L*	within
	BovineHD2700002886	rs42110372	27	10498902	0.10	3.11 × 10^−7^	*LOC112444670*	100 kb
	ARS-BFGL-NGS-116157	rs109601642	20	16333125	0.27	6.46 × 10^−7^	*LOC101907219*	100 kb
	ARS-BFGL-NGS-73835	rs41577664	15	80365537	0.48	7.43 × 10^−7^	*LOC112441589*	200 bp
	BovineHD1400023839	rs134726669	14	81745515	0.31	7.59 × 10^−7^	*MRPL13*	200 bp
Rear legs rear view	BovineHD0300015960	rs134130409	3	52695086	0.41	6.72 × 10^−8^	*BARHL2*	100 kb
	BovineHD4100014792	rs134139959	20	57471508	0.19	1.11 × 10^−7^	*FBXL7*	within
	Hapmap48798-BTA-51401	rs41638134	1	131310669	0.35	6.11 × 10^−7^	*LOC107132214*	20 kb
Rear legs side view	ARS-BFGL-NGS-91167	rs41565304	5	108222067	0.19	1.11 × 10^−9^	*ADIPOR2*	within
	BovineHD0300022142	rs43350216	3	76168946	0.38	4.05 × 10^−9^	*-*	-
	BovineHD1100001372	rs43656945	11	3761839	0.07	2.32 × 10^−7^	*INPP4A*	within
	BovineHD1100021155	rs136593856	11	74048825	0.44	5.07 × 10^−7^	*DNMT3A*	within
	BTB-01677645	rs42791722	10	52217450	0.09	7.40 × 10^−7^	*ALDH1A2*	within
	BTB-01518251	rs42639670	6	49588438	0.09	9.65 × 10^−7^	*PCDH7*	200 kb

**Table 4 animals-11-02259-t004:** KEGG analysis for the regional candidate genes with genome-wide significant association.

Term	Description	Gene Count	%	*p*-Value	Genes
bta00531:	Glycosaminoglycan degradation	3	2.9	0.00469386	*LOC527125*, *LOC509761*, *SPAM1*
bta01100:	Metabolic pathways	12	11.4	0.016110131	*INPP4A*, *LOC527125*, *LIPC*, *LOC520336*, *MGAT4A*, *ALDH1A2*, *DNMT3A*, *LOC100848875*, *LOC509761*, *POLE*, *GALNT9*, *SPAM1*
bta04020:	Calcium signaling pathway	4	3.8	0.052382966	*P2RX3*, *P2RX2*, *ADCY3*, *CACNA1C*
bta05414:	Dilated cardiomyopathy	3	2.9	0.056875544	*ADCY3*, *CACNA2D4*, *CACNA1C*
bta04916:	Melanogenesis	3	2.9	0.072824808	*POMC*, *WNT5B*, *DCY3*

## Data Availability

The data presented in this study are available on request from the corresponding author.

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
