# Peer review of "Genome-Wide Association Study Identifies Candidate Genes Associated with Feet and Leg Conformation Traits in Chinese Holstein Cattle"

_animals, 2021, doi:10.3390/ani11082259_

Round 1

Reviewer 1 Report

An important study contributing to genomic characterization of traits of economic importance in local cattle breeds. Authors need to improve on their communication to help the reader understand the study. I have offered a few suggestions in text to help the authors improve on the quality of the manuscript. In addition authors should update the literature to help improve on the discussion and conclusion of the study.

Reviewer 2 Report

This manuscript describes a GWAS analysis of feet and leg traits in Chinese Holstein cattle, followed by in-depth bioinformatics analysis of genetic architecture of those traits. The research presented is interesting and relevant. The analyses are complete and well described. The results are clearly presented and discussed.

The manuscript may benefit from providing additional det as listed below:

Line 95-96: How many classifiers participated in phenotypic data collection? Was it just one person, or many? If so, were they all equally trained?

Line 105: Add reference for SPSS software if available.

Line 110: Describe the effects of year, season, and parity. How many years (from … to); how was the season defined? How many parities were considered?

Line 124: Add reference for Plink software.

Line 153: Reference?

Lines 178-192: I could not see anything about the pedigree of the animals in your study, so I assume that GBLUP was used to estimate variance components (heritabilities and correlations), with genomic relationship matrix (G) replacing the pedigree relationship matrix A. If this is correct, please explain how G was constructed. Also, when talking about correlations, please discuss how they compare with correlations from other studies with similar traits (I would expect higher genetic correlations among the traits and consequently some SNPs affecting more than just one trait).

Line 209: The k value of 21 seems very high, and the cross validation error still seems NOT to decrease. Why? What happens if you go over 21? Also, the graph in Figure 2c should be rescaled with y-axis going from, e.g., 0.5 to 0.7, for the differences in CV error to be clearly visible. Regarding Figure 2b, I would recommend to order animals by the most (or the least) dominant cluster instead of showing the sequential order of animal IDs.

Minor edits:

The following comments relate only to spelling, grammar, or sentence structure in the manuscript, not to content:

  1. 70: … and mastitis …
  2. 71: …. Investigation, some studies have been conducted …
  3. 75: … in our study were to detect …
  4. 134: … in Chinese Holstein cattle herds …
  5. 209: ADMIXTURE (spelling).
  6. 284: The results of the Linkage Disequilibrium (LD) analysis indicate…
  7. 304: … small nucleolar RNA …
  8. 354: … major challenges …
  9. 365: … as covariates are considered effective ways …
  10. 366-367: … as MLM can miss … and lead …
  11. 370-371: … because of its advantage to completely control …
  12. 380: The figure 2a shows …
  13. 387: … analysis identified 20…
  14. 401: … plays a crucial role… (?)
  15. 407: … with conformation and reproductive performance traits …
  16. 413: … (INPP4A) reported association …
  17. 427: … (PCDH7) might be …
  18. 444-445: … Our findings suggest that …
